Predicting CoVID-19 community mortality risk using machine learning and development of an online prognostic tool

http://orcid.org/0000-0002-7110-9463 Das Ashis Kumar 1 adas8@worldbank.org
Mishra Shiba 2
Saraswathy Gopalan Saji 1
1 The World Bank , Washington, DC , USA
2 Credit Suisse Private Limited , Pune , India
Aly Sharif
Electronic publication date: 2020 Sep 28
Publication date: 2020
Volume: 8
Electronic Location ID: e10083
Received 2020 May 17; Accepted 2020 Sep 11
Copyright: © 2020 Das et al.
Copyright year: 2020
Copyright holder: Das et al.
License: This is an open access article distributed under the terms of the Creative Commons Attribution License, which permits unrestricted use, distribution, reproduction and adaptation in any medium and for any purpose provided that it is properly attributed. For attribution, the original author(s), title, publication source (PeerJ) and either DOI or URL of the article must be cited.
License URL: https://creativecommons.org/licenses/by/4.0/

Keywords: CoVID-19, Machine learning, Modelling, Mortality risk prediction, Decision support

Funding: The authors received no funding for this work.

==============================
Background

The recent pandemic of CoVID-19 has emerged as a threat to global health security. There are very few prognostic models on CoVID-19 using machine learning.

Objectives

To predict mortality among confirmed CoVID-19 patients in South Korea using machine learning and deploy the best performing algorithm as an open-source online prediction tool for decision-making.

Materials and Methods

Mortality for confirmed CoVID-19 patients (n = 3,524) between January 20, 2020 and May 30, 2020 was predicted using five machine learning algorithms (logistic regression, support vector machine, K nearest neighbor, random forest and gradient boosting). The performance of the algorithms was compared, and the best performing algorithm was deployed as an online prediction tool.

Results

The logistic regression algorithm was the best performer in terms of discrimination (area under ROC curve = 0.830), calibration (Matthews Correlation Coefficient = 0.433; Brier Score = 0.036) and. The best performing algorithm (logistic regression) was deployed as the online CoVID-19 Community Mortality Risk Prediction tool named CoCoMoRP (https://ashis-das.shinyapps.io/CoCoMoRP/).

Conclusions

We describe the development and deployment of an open-source machine learning tool to predict mortality risk among CoVID-19 confirmed patients using publicly available surveillance data. This tool can be utilized by potential stakeholders such as health providers and policymakers to triage patients at the community level in addition to other approaches.

Introduction

A novel coronavirus disease 2019 (CoVID-19) originated from Wuhan in China was reported to the World Health Organization in December of 2019 (WHO, 2020). Ever since, this novel coronavirus has spread to almost all major nations in the world resulting in a major pandemic. As of June 08, 2020, it has contributed to more than 7 million confirmed cases and about 404,000 deaths (Coronavirus Resource Center, 2020). The first CoVID-19 case was diagnosed in South Korea on January 20, 2020. According to the Korea Centers for Disease Control and Prevention (KCDC), there have been 11,814 confirmed cases and 273 deaths due to CoVID-19 as of June 08, 2020 (KCDC, 2020).

In the field of healthcare, accurate prognosis is essential for efficient management of patients while prioritizing care to the more needy. In order to aid in prognosis, several prediction models have been developed using various methods and tools including machine learning (Chen & Asch, 2017; Qu et al., 2019; Lei et al., 2020). Machine learning is a field of artificial intelligence where computers simulate the processes of human intelligence and can synthesize complex information from huge data sources in a short period of time (Benke & Benke, 2018). Though there have been a few prediction tools on CoVID-19, only a handful have utilized machine learning (Wynants et al., 2020). To the best of our knowledge, by far there is no publicly available online CoVID-19 prognosis prediction tool from the general population of confirmed cases using machine learning. We attempt to apply machine learning on the publicly available CoVID-19 data at the community level from South Korea to predict mortality.

Our study had two objectives: (1) predict mortality among confirmed CoVID-19 patients in South Korea using machine learning algorithms, and (2) deploy the best performing algorithm as an open-source online prediction tool for decision-making.

Materials and Methods

Patients

Patients for this study were selected from the data shared by Korea Centers for Disease Control and Prevention (KCDC, 2020). The timeframe of this study was from the beginning of the detection of the first case (January 20, 2020) through May 30, 2020. Though there have been 11,814 confirmed cases according to the KCDC by this date, there were only a total of 4,004 patients in the publicly available dataset. Our inclusion criteria were confirmed CoVID-19 cases with availability of demographic, exposure and diagnosis confirmation features along with the outcome. We excluded patients those had missing features—sex (n = 330) and age (n = 150), and thus, 3,524 patients were included in the final analysis.

Outcome variable

The outcome variable was mortality and it had a binary distribution—“yes” if the patient died, or “no” otherwise.

Predictors

The predictors were individual patient level demographic and exposure features. They were four predictors: age group, sex, province, and exposure. There were ten age groups as follows below 10 years, 10–19 years, 20–29 years, 30–39 years, 40–49 years, 50–59 years, 60–69 years, 70–79 years, 80–89 years, 90 years and above. Patients represented all 17 provinces of South Korea (Busan, Chungcheongbuk-do, Chungcheongnam-do, Daegu, Daejeon, Gangwon-do, Gwangju, Gyeonggi-do, Gyeongsangbuk-do, Gyeongsangnam-do, Incheon, Jeju-do, Jeollabuk-do, Jeollanam-do, Sejong, Seoul, and Ulsan). Patients were exposed in several settings, such as nursing home, hospital, religious gathering, call center, community center, shelter and apartment, gym facility, overseas inflow, contact with patients and others.

Statistical methods

Descriptive analysis

We performed descriptive analyses of the predictors by respective stratification groups and present the results as numbers and proportions. Potential correlations between predictors were tested with Pearson’s correlation coefficient.

Predictive analysis

We applied machine learning algorithms to predict mortality among CoVID-19 confirmed cases. Machine learning is a branch of artificial intelligence where computer systems can learn from available data and identify patterns with minimal human intervention (Deo, 2015). Typically, in machine learning several algorithms are tested on data and performance metrics are used to select the best performing algorithm. While selecting the algorithms, we considered commonly used machine learning algorithms in healthcare research that have lower training time as well as lower lag time when built into an online application. Thus, the selected algorithms were—logistic regression, support vector machine, K neighbor classification, random forest and gradient boosting. Using grid search function, we also performed hyperparameter tuning (i.e., selection of the best parameters) for each algorithm (Table S1). Logistic regression is best suited for a binary or categorical output. It tries to describe the relationship between the output and predictor variables (Jiang et al., 2017). In support vector machine (SVM) algorithm, the data is classified into two classes based on the output variable over a hyperplane (Jiang et al., 2017). The algorithm tries to increase the distance between the hyperplane and the most proximal two data points in each class. SVM uses a set of mathematical functions called kernels, which transform the inputs to required forms. In our SVM algorithm, we used a radial kernel. K Nearest Neighbors (KNN) is a non-parametric approach that decides the output classification by the majority class among its neighbors (Raeisi Shahraki, Pourahmad & Zare, 2017). The number of neighbors can be altered to arrive at the best fitting KNN model. Random forest algorithm uses a combination of decision trees (Rigatti, 2017). Decision trees are generated by recursively partitioning the predictors. New attributes are sequentially fitted to predict the output. Gradient boosting (GB) algorithm uses a combination of decision trees (Natekin & Knoll, 2013). Each decision tree dynamically learns from its precursor and passes on the improved function to the following. Finally, the weighted combination of these trees provides the prediction. A decision tree’s learning from the precursor and the number of subsequent trees can be respectively adjusted using learning rate and number of trees parameters.

Evaluation of the performance of the algorithms

We split the data into training (80%) and test cohorts (20%). Initially, the algorithms were trained on the training cohort and then were validated on the test cohort (new data) for determining predictions. The data was passed through a 10-fold cross validation where the data was split into training and test cohorts at 80/20 ratio randomly ten times. The final prediction came out of the cross-validated estimate. As our data was imbalanced (only 2.1% output were with the condition against 97.9% without), we applied two oversampling techniques called synthetic minority oversampling technique (SMOTE) and adaptive synthetic (ADASYN) method to enhance the learning on the training data (Chawla et al., 2002; Nnamoko & Korkontzelos, 2020). SMOTE creates synthetic samples from the minority class (cases with deaths in our data) according to feature space similarities between nearest neighbors (Chawla et al., 2002). ADASYN adaptively generates synthetic samples based on their difficulty in learning (He et al., 2008).

The performance of the algorithms were evaluated for discrimination, calibration and overall performance. Discrimination is the abillity of the algorithm to separate out patients with the mortality risk from those without, where as calibration is the agreement between observed and predicted risk of mortality. An ideal model should have the best of both discrimination and calibration. We tested discrimination with area under the receiver operating characteristics curve (AUC) and calibration with Matthews correlation coefficient. A receiver operator characteristic (ROC) curve plots the true positive rate on y-axis against the false positive rate on x-axis (Huang et al., 2020). AUC is score that measures the area under the ROC curve and it ranges from 0.50 to 1.0 with higher values meaning higher discrimination. Matthews correlation coefficient (MCC) is a measure that takes into account all four predictive classes—true positive, true negative, false positive and false negative (Chicco & Jurman, 2020). Brier score simultaneously account for discrimination and calibration (Huang et al., 2020). A smaller Brier score indicates better performance. We also estimated accuracy, sensitivity and specificity. Accuracy is a measure of correct classification of death cases as death and survived cases as survived (Huang et al., 2020). Sensitivity is a measure of correctly predicting death among all those who died, whereas specificity is a measure of correctly predicting survival among all those who survived. In addition, relative influence of the predictors with the output was estimated using the random forest (mean decrease Gini coefficients—MDG) and logistic regression algorithm (regression coefficients) (Xie & Coggeshall, 2010). MDG quantifies which predictor contributed most to the classification accuracy.

The statistical analyses were performed using Stata Version 15 (StataCorp LLC, College Station, TX, USA), Python programing language Version 3.7.1 (Python Software Foundation, Wilmington, DE, USA); e1071 and caret packages of R programming language Version 3.6.3 (R Foundation for Statistical Computing, Vienna, Austria). The web application was built using the Shiny package for R and deployed with Shiny server.

Results

Patient profile

The profile of the patients is presented in Table 1. Out of 3,524 confirmed patients, a slightly more than half were females (55.1%). Among the age groups, the maximum patients were from 20 to 29 years (24.4%), followed by 50–59 years (17.7%), 30–39 years (14%), 40–49 years (13.7%), and 60–69 years (12%). Gyeongsangbuk-do (35.1%), Gyeonggi-do (23.5%) and Seoul (16%) provinces together presented the maximum patients. Considering the source/mode of infection, the largest group had unknown mode (39.3%) followed by direct contact with patients (29.8%) and from overseas (17.4%). According to this available data source, there were 74 deaths accounting for 2.1% of the patients.

Table 1 Sample characteristics.

Variable	Number	Proportion (%)	
Sex			
Female	1,940	55.1	
Male	1,584	45.0	
Age group (years)			
Below 10	60	1.7	
10–19	160	4.5	
20–29	859	24.4	
30–39	494	14.0	
40–49	483	13.7	
50–59	625	17.7	
60–69	423	12.0	
70–79	210	6.0	
80–89	162	4.6	
90 and above	48	1.4	
Province			
Busan	144	4.1	
Chungcheongbuk-do	52	1.5	
Chungcheongnam-do	146	4.1	
Daegu	63	1.8	
Daejeon	46	1.3	
Gangwon-do	52	1.5	
Gwangju	30	0.9	
Gyeonggi-do	829	23.5	
Gyeongsangbuk-do	1,236	35.1	
Gyeongsangnam-do	119	3.4	
Incheon	92	2.6	
Jeju-do	14	0.4	
Jeollabuk-do	20	0.6	
Jeollanam-do	19	0.5	
Sejong	47	1.3	
Seoul	563	16.0	
Ulsan	52	1.5	
Exposure			
Nursing home	46	1.3	
Hospital	37	1.1	
Religious gathering	160	4.5	
Call center	135	3.8	
Community center, shelter and apartment	68	1.9	
Gym facility	34	1.0	
Overseas inflow	612	17.4	
Contact with patients	1,049	29.8	
Others	1,383	39.3	
Outcome			
Survived	3,450	97.9	
Died	74	2.1	
Total	3,524	100	

The correlation coefficients among the predictors ranged from −0.12 to 0.22. Using the random forest algorithm, we estimated the relative influence of the predictors (Fig. 1). According to the random forest algorithm, age was the most important predictor followed by exposure, sex and province, whereas this order was sex, age, province and exposure as per logistic regression

Figure 1 Relative importance of predictors.

(A) Random forest, (B) Logistic regression.

Performance of the algorithms

Table 2 presents the performance metrics of all algorithms—logistic regression, support vector machine, K nearest neighbor, random forest and gradient boosting. The area under receiver operating characteristic curve (AUC) ranged from 0.644 to 0.830 with the best score for the logistic regression (SMOTE) algorithm. Similarly, logistic regression (SMOTE) performed the best on Matthews correlation coefficient. It was in the middle for the performance on Brier score. The accuracy of all algorithms was very similar with random forest (SMOTE) performing the best (0.972) and K nearest neighbor with the least score (0.924). Considering all the performance metrics, logistic regression (SMOTE) was the best performing algorithm.

Table 2 Performance of the machine learning algorithms.

Algorithm	Oversampling method	Area under
ROC curve	Matthews correlation coefficient	Brier score	Sensitivity	Specificity	Accuracy	
Logistic regression	SMOTE#	0.830	0.433	0.036	0.692	0.968	0.965	
ADASYN*	0.823	0.376	0.049	0.692	0.955	0.968	
Support vector machine	SMOTE#	0.825	0.393	0.045	0.692	0.959	0.970	
ADASYN*	0.786	0.345	0.048	0.615	0.958	0.971	
K nearest neighbor	SMOTE#	0.644	0.253	0.031	0.307	0.981	0.942	
ADASYN*	0.759	0.410	0.028	0.538	0.979	0.924	
Random forest	SMOTE#	0.787	0.351	0.046	0.615	0.959	0.972	
ADASYN*	0.787	0.351	0.046	0.615	0.959	0.971	
Gradient boosting	SMOTE#	0.787	0.351	0.046	0.615	0.959	0.971	
ADASYN*	0.787	0.351	0.046	0.615	0.959	0.971	
Notes:

# SMOTE, Synthetic minor oversampling technique.

* ADASYN, Adaptive synthetic sampling.

Online CoVID-19 mortality risk prediction tool—CoCoMoRP

The best performing model—logistic regression (SMOTE) was deployed as the online mortality risk prediction tool named as “CoVID-19 Community Mortality Risk Prediction”—“CoCoMoRP” (https://ashis-das.shinyapps.io/CoCoMoRP/). Figure 2 presents the user interface of the prediction tool. The web application is optimized to be conveniently used on multiple devices such as desktops, tablets, and smartphones.

Figure 2 CoCoMORP online CoVID-19 community mortality risk prediction tool.

The user interface has four boxes to select input features as drop-down menus. The features are sex (two options—male and female), age (ten options—below 10 years, 10–19 years, 20–29 years, 30–39 years, 40–49 years, 50–59 years, 60–69 years, 70–79 years, 80–89 years, 90 years and above), province (all 17 provinces—Busan, Chungcheongbuk-do, Chungcheongnam-do, Daegu, Daejeon, Gangwon-do, Gwangju, Gyeonggi-do, Gyeongsangbuk-do, Gyeongsangnam-do, Incheon, Jeju-do, Jeollabuk-do, Jeollanam-do, Sejong, Seoul, Ulsan), and exposure (nine options—nursing home; hospital; religious gathering; call center; community center, shelter and apartment; gym facility; overseas inflow; contact with patients; and others).

The user has to select one option each from the input feature boxes and click the submit button to estimate the CoVID-19 mortality risk probability in percentages. For instance, the tool gives a CoVID-19 mortality risk prediction of 94.1% for a male patient aged between 80 and 89 years from Seoul province coming in contact with patient as the exposure.

Discussion

The CoVID-19 pandemic is a threat to global health and economic security. Recent evidence for this new disease is still evolving on various clinical and socio-demographic dimensions (Sun et al., 2020; Chen et al., 2020; Li et al., 2020). Simultaneously, health systems across the world are constrained with resources to efficiently deal with this pandemic. We describe the development and deployment of an open-source artificial intelligence informed prognostic tool to predict mortality risk among CoVID-19 confirmed patients using publicly available surveillance data. This tool can be utilized by potential stakeholders such as health providers and policy makers to triage patients at the community level in addition to other approaches.

There are a few online predictive applications on CoVID-19. A web-application was developed in China from hospital admissions in a single hospital to identify suspected CoVID-19 cases (Feng et al., 2020). The study used patient demographics, vital signs, blood examinations, clinical signs and symptoms and infection-related biomarkers. The application used four different algorithms—logistic regression with LASSO, logistic regression with Ridge regularization, decision tree, and Adaboost. LASSO regularized logistic regression was the best performer with an AUC of 0.8409. Another web application uses hospitalization data from China, Italy and Belgium to predict severity of illness (Wu et al., 2020). With the support of a machine-learning model, this application assesses severity risk for CoVID-19 patients at hospital admission. Clinical, laboratory, and radiological characteristics were the predictors. Using logistic regression, the AUCs ranged from 0.84 to 0.89 and accuracies ranged from 74.4% to 87.5%. Another application was developed in Italy from patients’ demographic and blood test parameters (Brinati et al., 2020). This application used six classifiers—decision tree, extremely randomized trees, random forest, three-way random forest, KNN, SVM, logistic regression and naïve bias. Random forest was the best performing algorithm with an accuracy of 82%. Similar to the two available online predictive applications on CoVID-19, our study also found logistic regression as the best performing algorithm. The AUC of the best performing algorithm in our study (0.83) is similar to that of other applications as well. However, the accuracy of our best performing algorithm (96%) is the highest when compared with similar CoVID-19 online applications. Nevertheless, there are two main differences between our and other online applications. First, our study uses only community level demographic features that are publicly available. Secondly, the sample in our study comprises of community dwellers whereas in others they come from hospitalized patients.

Using separate methodologies (regression coefficients for logistic regression and mean decrease Gini coefficient), we ranked the predictors for their contribution towards the classification accuracy of the outcome in our study. Similar to the current evidence, age was the first and second important feature for random forest and logistic regression algorithms respectively (Huang et al., 2020; Wang et al., 2020). Coexistence of chronic illnesses at old age might be related to higher mortality. There were differences in the rankings for other predictors as well which could be attributed to the differences in the methodologies employed to rank the predictors.

Our study has several strengths. First, to the best of our knowledge, our CoVID-19 community mortality risk prediction study is the first of its kind that uses artificial intelligence tools. Secondly, we developed the prediction model using simple and readily available data by a public health agency. Finally, our risk prediction tool is publicly available for estimating the community mortality risk due to CoVID-19.

One major limitation of this tool is unavailability of crucial clinical information on symptoms, risk factors and clinical parameters. Recent research has identified certain symptoms, preexisting illnesses and clinical parameters as strong predictors of prognosis and severity of progression for CoVID-19 (Li et al., 2019, 2020; Guan et al., 2020). These crucial pieces of information are not publicly available so far in the surveillance data, so the tool could not be tested to include these features. Inclusion of these additional features may improve the reliability and relevance of the tool. Therefore, we urge the users to balance the predictions from this tool against their own and/or health provider’s clinical expertise and other relevant clinical information. Secondly, we did not use a held-out subset of data for validation that was not included in the cross-validation process. This might have led to overfitting of the models with the available data. The third limitation pertains to lack of availability of the complete data. According to the reports, there were 11,814 confirmed cases and 273 deaths (case fatality rate 2.3%) due to CoVID-19 in South Korea as of June 08, 2020. However, our analysis using the publicly released database found 3,529 cases and 74 deaths (case fatality rate 2.1%) until May 30, 2020. Though the case fatality rates are similar, our analysis uses respectively about a third and a fourth of totally reported cases and deaths. As more data are released publicly, we would continue to update our analyses and the web-application. However, we strongly believe that more deidentified data and patient clinical features should be made available by the public health entities in a pandemic situation like CoVID-19. Emerging evidence suggests strong associations of CoVID-19 severity with patient clinical features such as vitals at hospital admission (temperature, blood pressure, respiration rate, and oxygen saturation), blood test parameters (complete blood count, liver and renal function tests), and preexisting conditions (diabetes, hypertension, cardiovascular and renal diseases, and chronic obstructive pulmonary disease) (Huang et al., 2020; Guan et al., 2020; Li et al., 2020). Inclusion of such patient level clinical features in the publicly available databases would enable development of more robust and relevant clinical decision support applications. Moreover, the publicly available data presents only a fourth of the total confirmed cases in the country. Release of more complete data would aid in proper optimization of the web application that reflects the true nature of the burden.

Conclusions

We tested multiple machine learning models to accurately predict deaths due to CoVID-19 among confirmed community cases in the Republic of Korea. Using the best performing algorithm, we developed and deployed an online mortality risk prediction tool. Our tool offers an additional approach to informing decision making for CoVID-19 patients.

Supplemental Information

Supplemental Information 1 Hyperparameter tuning of machine learning algorithms using grid search.

Click here for additional data file.

We are grateful to Korea Center for Disease Control and Prevention for making this data publicly available. The views expressed in the paper are that of the authors and do not reflect that of their affiliations. This particular work was conducted outside of the authors’ organizational affiliations.

Additional Information and Declarations

Competing Interests

Author Contributions

Data Availability

Ashis Kumar Das and Saji Saraswathy Gopalan are employed by the World Bank. Shiba Mishra is employed by Credit Suisse Private Limited.

Ashis Kumar Das conceived and designed the experiments, performed the experiments, analyzed the data, prepared figures and/or tables, authored or reviewed drafts of the paper, and approved the final draft.

Shiba Mishra conceived and designed the experiments, performed the experiments, analyzed the data, prepared figures and/or tables, authored or reviewed drafts of the paper, and approved the final draft.

Saji Saraswathy Gopalan conceived and designed the experiments, performed the experiments, prepared figures and/or tables, authored or reviewed drafts of the paper, and approved the final draft.

The following information was supplied regarding data availability:

The codes are available at GitHub: https://github.com/ShibaMis/Data-Analysis.

The data used in this study is made publicly available by the Korean Centers for Disease Control and Prevention at Kaggle. This requires a free Kaggle account.

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
