# Peer review of "Predicting CoVID-19 community mortality risk using machine learning and development of an online prognostic tool"

_PeerJ, doi:10.7717/peerj.10083_

## Round 0.1 · original submission · Major Revisions

Employing the ML algorithms you contrast and using cross-validation is commendable. However, in addition to the reviewer comments, the manuscript's results currently do not meet the publication bar. Your dataset could have included the month of May, providing 10,900 cases and over 200 deaths instead of the current 3999 cases and 66 deaths (the main outcome) which is too few to determine the important diagnostic test accuracy estimates missing in the manuscript, sensitivity and specificity. How sensitive and specific the online tool is vital to educating the public and interested users as to the accuracy of this tool. Please consider adding the additional month, which should be available in your dataset you extracted, otherwise justify how can the accuracy estimates you report be as accurate as you claim.

Furthermore, the dataset needs to be included with the manuscript's next revision to allow readers to be able to replicate your work. Once these issues are addressed and the reviewer comments there is a very high chance the manuscript would be accepted.

Editor comments:
Line 85: You report collecting data on socio-economic status, what measures did you use, also no SES results are presented in the results
Line 92: What are the others (even if few list them please)
Lines 110, 111: several short sentences that do not connect, please reword for clarity
Lines 126: not clear if the k-fold cross validation was based on 80% and the accuracy parameters were on the 20%? It is paramount that the accuracy estimates be based on a new subset that was not involved in the cross-validation, please clarify
Line 139-140: Thanks for defining accuracy but this is overall accuracy, this is only of limited value, we need to know Sensitivity and Specificity since you are dealing with mortality, as a health outcome these will be important.
Line 149: Please name the R packages

·

Basic reporting

The study has two laudable objectives: predict mortality among confirmed CoVID-19 patients in South Korea using machine learning algorithms, and deploy the best performing algorithm as an open-source online prediction tool for decision-making. They employ a reasonable yet imbalanced dataset of 3,299 patients and a suit of socio-demographic and exposure features, with mortality being the outcome variable. I believe the study has merit and is clearly timely.

Experimental design

I detected the following major methodological issues:

- Predictive accuracy will be greatly affected by hyper-parameter choice. Yet, the authors do not mention how hyper-parameters were tuned.

- The dataset is highly imbalanced (2.1% mortality), so I suggest using and comparing multiple oversampling approaches (authors use only one).

Validity of the findings

Most methods seem appropriate and results robust. However, I urge the authors to assess how hyper-parameter tuning and oversampling approaches influence their predictive accuracy.
Additionally, I suggest the authors discuss the unaccounted influece of spatio-temporal processess, and provide an additional figure showing a map of all provinces and the number of infected cases in each by month.

Additional comments

L62-66: A quick search in medXiv using the terms “covid-19 and achine learning” yields 379 results, so this is not true. Please clarify what you did differently from all these previous studies.

L103-105: Justify your choice of algorithms, saying that they are commonly used is not enough.

L103-L122: You should explain how you performed hyper-parameter tuning and not simply report hyper-parameter values. Mention if you used an adaptive approach or a grid, and in the later case provide the range of values. Predictive accuracy will be greatly affected by hyper-parameter choice, so this section needs to be thoroughly revised.

L128-L130: As data is highly imbalanced (only 2.1% output were with the condition against 97.9% without), I suggest using and comparing multiple oversampling approaches, not only SMOTE.

L194: Regions with more cases are likely to infect neighboring regions faster than more distant regions. And perhaps mortality risk is related to the number of infections/strains a person is exposed to. Your algorithms are not accounting for this (they are not spatially explicit), nor they incorporate time since infection. I suggest you mention this in your discussion, and provide an additional figure showing a map of all provinces and the number of infected cases in each by month.

Reviewer 2 ·

Basic reporting

In this paper, the authors propose to predict CoVID-19 community morality and developed a web portal to showcase the prediction results.

Experimental design

The authors adopted 5 machine learning algorithms and tested on a CoVID-19 dataset from Korea.

Validity of the findings

The results demonstrate the effectiveness of the prediction models (e.g., AUC = 0.886).

Additional comments

1) Since the data is highly imbalanced (97.9% negative, as shown in Table 1), accuracy should not be used as an evaluation metrics. The major metric should be AUC.

2) Among the 5 classifiers, the features importances are only showed for Random Forest. It would be better to show the results from all classifiers, especially Logistic Regression as it provides p-values.

3) Minor: it would be more clear if the total number of predictors are explicitly shown in the paper.

---

## Round 0.2 · Major Revisions

Thank you for responding to comments from the reviewers myself. Please respond to Reviewer 2's comment regarding presenting the logistic regression odds ratios. I also wanted to follow up on some of my comments that either need further clarification or should be addressed:

1) In my earlier request to include the month of May, I based it on your first submission's statement that "According to the Korea Centers for Disease Control and Prevention (KCDC), there have been 10,909 confirmed cases 56 and 256 deaths due to CoVID-19 as of May 11, 2020." but based on the very modest increase in cases and deaths is it fair to assume that the database you are accessing data from is not up to date with Korea CDC estimates? (since you did not gain the thousands that could have potentially increased your sample size?)
If that is the case please add a sentence in your data so readers do not wonder why the discrepancy in state reported cases and your data.

2) With regards to cross-validation, I am familiar with it thank you, but the question I had was whether the testing was on a subset that was never used in the "learning" in any of the K subsets of the cross validation. For this to happen, I think you would have had to create 3 subsets of the data (learn/validate and test) not 2 (learn and test) as you described. Notice that the learn and validate are what you have in K-fold cross validation, but the 20% would never be used until that third stage of testing the model accuracy as in Fig 3b from Lever, J. K. M., and N. Altman. 2016. Model selection and over fitting; Nat. Methods 13:703–704. (Otherwise your response still describes using test data from each of the 10 subsets, meaning at some point some of this test data is recycled after being used for model learning, albeit in a different K). Of course other variants exist in cross-validation so feel free to describe with more detail how you protected a subset (10-20% of the data usually) to test and estimate your model accuracy among other parameters, and that was not used in any subset of the data.

Finally your discussion is very thin, at 27 double spaced lines this amounts to less than a single column paragraph in print font when you could and should add more discussion. For example (if I may help at all), you don't contrast any other tools even broadly across other diseases, accuracy of data, lag in data collection, role of prevention and mitigation in influencing your model prediction; if Korea or other nations stop prevention and control do you think your estimates will be the same; under or overestimate the observed mortalities); limitations in not including more data from the individual - basically what data should public health entities consider recording also so future models like this be stronger; what about the strengths of the study in using simple readily available data, what about the caveats for in accuracy especially false positives (with the positive status being a prediction for mortality) and what factors affect false positives (specificity basically).

Please address these and your submission will have a great chance of being accepted.

·

Basic reporting

Fine

Experimental design

Fine

Validity of the findings

Fine

Additional comments

I am happy with the improvements made in the revised version of the manuscript.

Reviewer 2 ·

Basic reporting

N/A

Experimental design

N/A

Validity of the findings

N/A

Additional comments

In this revision, the authors have addressed my previous 2 suggestions. Yet for the feature importance, I still recommend at least report the odd ratio and p-value of Logistic Regression, which is a very common practice in predictive modeling and also one of the important benefit to adopt this specific algorithm. Also, this provide an additional result because then the features can be compared based on their importance derived from both Random Forest and Logistic Regression.

Thank you very much!

---

## Round 0.3 · accepted · Accept

Thank you for addressing the reviewers' and my comments. I am pleased to inform you that your manuscript has been accepted for publication. Congratulations!
Best wishes,
Sharif